# Effective Penetration of a Liposomal Formulation of Bleomycin through Ex-Vivo Skin Explants from Two Different Species

**DOI:** 10.3390/cancers14041083

**Published:** 2022-02-21

**Authors:** Giulia Ferrari, Lisa Y. Pang, Fabio De Moliner, Marc Vendrell, Richard J. M. Reardon, Andrew J. Higgins, Sunil Chopra, David J. Argyle

**Affiliations:** 1Roslin Institute, The Royal (Dick) School of Veterinary Studies, University of Edinburgh, Edinburgh EH25 9RG, UK; giuliaferrari1988@gmail.com (G.F.); richard.reardon@ed.ac.uk (R.J.M.R.); david.argyle@ed.ac.uk (D.J.A.); 2Centre for Inflammation Research, The Queen’s Medical Research Institute, University of Edinburgh, Edinburgh EH16 4TJ, UK; fdemoli@exseed.ed.ac.uk (F.D.M.); marc.vendrell@ed.ac.uk (M.V.); 3The London Dermatology Centre, London W1G 8AS, UK; ajhvet@gmail.com (A.J.H.); sunil.chopra3@btinternet.com (S.C.)

**Keywords:** liposomes, Bleosome, Bodipy-FL, topical chemotherapy, skin penetration

## Abstract

**Simple Summary:**

Bleomycin, a chemotherapy drug, is currently injected into patients, but this can damage healthy tissues. Ideally, we would like to apply bleomycin directly onto a skin tumour but bleomycin is a big molecule and cannot pass through the skin or directly enter into cancer cells to kill them. Therefore, we need to find new ways of packaging the drug to get it inside cancer cells. Liposomes are small artificial bubbles made of from the same building blocks as our skin and cell membranes that can be filled with pharmaceutical drugs. In this study we propose that liposomes can assist with the delivery of bleomycin by improving penetration through the skin. We are using a new compound called Bleosome, which contains liposomes packed with bleomycin. We found that Bleosome penetrated more through the healthy skin of dogs and horses than bleomycin. These are promising results, indicating that Bleosome may be an effective treatment, with easy application and limited side-effects, to treat skin cancer.

**Abstract:**

Bleomycin is a chemotherapy agent that, when administered systemically, can cause severe pulmonary toxicity. Bleosome is a novel formulation of bleomycin encapsulated in ultra-deformable (UD) liposomes that may be applicable as a topical chemotherapy for diseases such as non-melanoma skin cancer. To date, the ability of Bleosome to effectively penetrate through the skin has not been evaluated. In this study, we investigated the ability of Bleosome to penetrate through ex vivo skin explants from dogs and horses. We visualized the penetration of UD liposomes through the skin by transmission electron microscopy. However, to effectively image the drug itself we fluorescently labeled bleomycin prior to encapsulation within liposomes and utilized multiphoton microscopy. We showed that UD liposomes do not penetrate beyond the stratum corneum, whereas bleomycin is released from UD liposomes and can penetrate to the deeper layers of the epidermis. This is the first study to show that Bleosome can effectively penetrate through the skin. We speculate that UD liposomes are penetration enhancers in that UD liposomes carry bleomycin through the outer skin to the stratum corneum and then release the drug, allowing diffusion into the deeper layers. Our results are comparative in dogs and horses and warrant further studies on the efficacy of Bleosome as topical treatment.

## 1. Introduction

In both veterinary and human medicine there is an unmet clinical need for effective anti-cancer drugs that can be applied topically. Notably, the standard treatment of non-melanoma skin cancers (NMSC), including basal cell carcinoma and squamous cell carcinoma, are surgical modalities that can cause severe pain, inflammation, and scarring [1,2]. These side-effects can be significant given that the tumors may be spread over large areas of the body, therefore anti-cancer drugs that can be applied topically would be therapeutically and cosmetically valuable by reducing surgical costs and by avoiding undesirable scarring. However, a major challenge in developing these non-invasive modalities is the degree of penetration of the drug through the skin in sufficient quantities to target cancer cells [3].

Bleomycin belongs to a family of glycopeptide antibiotics and is used primarily as a cytotoxic chemotherapy drug. In human medicine it is an effective treatment against squamous cell carcinoma, malignant lymphomas, testicular cancers, germinal cell tumors, AIDS-associated Kaposi sarcoma and osteosarcoma [4], and is commonly used as part of combination chemotherapy regimens, as its toxicity does not overlap with that of other anticancer drugs [5]. The proposed mechanism of action of bleomycin is the inhibition of DNA synthesis by causing DNA single and double strand breaks [4]. At high concentrations, cellular RNA and protein synthesis are also suppressed [6,7]. Generally, bleomycin administration is via the parenteral route, as oral bioavailability is limited [8]. Bleomycin toxicity is dose-dependent and the most common adverse side-effect is pulmonary inflammation, which has been reported to develop into fatal lung fibrosis in 1% to 3% of patients [9]. In the case of NMSCs, topical application of bleomycin may avoid these systemic side-effects, but because bleomycin itself is a large hydrophilic molecule with a polar charge, it is unable to penetrate lipophilic skin barriers and efficiently diffuse across the plasma membrane to reach its site of action [4,10]. To overcome these issues, bleomycin has been administered by electrochemotherapy, which uses short electric impulses to transiently permeabilize the cell membrane allowing delivery of non-permeant drugs to the interior of the cell. This method has been successful in the management of skin tumors in both humans [11] and companion animals [12], and does reduce the major systemic side-effects of bleomycin but can also cause pain, muscle contractions, and discoloration of the area treated. Patients also generally undergo a general or local anesthetic, which, especially in veterinary patients, can be distressing for patients and owners.

In this study we have utilized a novel formulation of bleomycin called Bleosome, which negates a number of adverse side-effects of bleomycin administered systemically or by electrochemotherapy and does not require either local or general anesthetic. Bleosome consists of bleomycin encapsulated within ultra-deformable (UD) liposomes [13]. Liposomes have been heralded as ‘magic bullets’, as they are able to deliver a chemotherapy payload directly into cancer cells [14]. Liposomes have a strong tendency to form in aqueous solutions and are composed of spherical phospholipid bilayers that enclose an aqueous internal compartment which can be loaded with desired cargo [15]. The proposed benefits of carrier-mediated drugs include greater solubility, longer duration of exposure, selective delivery of the drug to the site of action, enhanced therapeutic index and the potential to overcome resistance associated with the free-form of the drug. Unfortunately, conventional liposomes cannot effectively penetrate through the skin barrier due to their rigidity [16]. However, UD liposomes, which include an edge activator, usually a single chain surfactant added to the phospholipid bilayer, can pass through the skin via pores that are much smaller in diameter than their own diameter by undergoing rapid changes in shape and fusing with lipids [17]. Skin is a complex organ consisting of a stratified cellular epidermis, an underlying dermis of connective tissue, and beneath the dermis there is a layer of subcutaneous fat separated from the rest of the body by a vestigial layer of striated muscle [18]. Beyond initial penetration of the skin, the mechanism by which UD liposomes improve drug delivery is yet to be fully elucidated. Two mechanisms have been proposed: (1) UD liposomes act as drug carrier systems, where intact UD liposomes enter the stratum corneum carrying the encapsulated drug deeper into the skin [19]; or (2) UD liposomes act as penetration enhancers, where liposome bilayers enter the stratum corneum and modify the intracellular lipid lamellae, releasing the encapsulated drug and enabling the penetration of the free drug deeper into the skin [20,21]. Here we show that Bleosome can penetrate further through ex vivo skin explants than free bleomycin in both canine and equine species. To determine the mechanism by which UD liposomes improve drug delivery, we utilized a combination of transmission electron microscopy (TEM) and multiphoton microscopy (MP) to visualize the UD liposomes and fluorescently labeled drug, respectively. We found that UD liposomes act as penetration enhancers: liposomes are visible throughout stratum corneum but not beyond, whereas the fluorescently labeled drug is seen within the stratum corneum as dense spots as the drug is enclosed with the liposome, but in the deeper layers of the epidermis the drug is more diffuse, indicating that it has been released from the UD liposome. This is the first study to show that this novel formulation of bleomycin can effectively penetrate through the skin of dogs and horses highlighting its clinical potential.

## 2. Materials and Methods

### 2.1. Cell Culture

A canine melanoma cell line, CML10 (a kind gift from Dr Lauren Wolfe, Auburn University) was grown in Dulbecco’s modified Eagle’s medium (DMEM) low glucose supplemented with 10% foetal bovine serum (FBS) and 1% penicillin/streptomycin (ThermoFisher Scientific, Waltham, MA, USA). A canine mastocytoma cell line, C2 (a kind gift from Dr Richard Elders, University of Edinburgh) was grown is Eagle DMEM (ThermoFisher Scientific) supplemented with 5% FBS, 1% non-essential amino acids, 1% Glutamax and 25 mg of gentamicin. All cell cultures were maintained at 37 °C in a humidified 5% CO_2_ incubator.

### 2.2. Stock Solutions of Bleosome and Bleomycin

Bleosome and bleomycin were provided at 0.2% in a cream. Both formulations were manufactured by Ascot Laboratories Ltd. (London, UK) and were a kind gift from SPS Animal Care Ltd. (London, UK). All compounds were stored at 4 °C.

### 2.3. Cell Viability

Cells were seeded in triplicate in opaque 96-well plates (Corning, Glendale, AZ, USA) at 500 cells/well. Cells were incubated 37 °C, 5% CO_2_ for 24 h prior to being treated with the indicated dose of Bleosome or bleomycin. Cell viability was assayed 48 h after treatment using the CellTiterGlo^®^ Luminescent Cell Viability Assay (Promega, Madison, WI, USA) according to the manufacturer’s instructions. Luminescence was recorded using a Viktor3 luminometer (PerkinElmer, Waltham, MA, USA). Data were averaged and then normalized against the average signal of untreated/negative control samples.

### 2.4. Fluorescent Labelling of Bleomycin

We coupled bleomycin sulfate (A2–B2) (120 mg) to the green fluorophore boron-dipyrromethene (BODIPY-FL). This reaction was carried out twice: (1) at a ratio of 0.7 BODIPY-FL NHS (8.1 mg): 1 bleomycin sulfate (45 mg); and (2) at a ratio of 0.8 BODIPY-FL NHS (14.6 mg): 1 bleomycin sulfate (71 mg). Both compounds were dissolved in DMSO, then BODIPY-FL NHS solution (~5 mg/mL) was slowly added dropwise to the bleomycin solution (~20 mg/mL) over 30 min with a Pasteur pipette. The coupling reaction was monitored by high-performance liquid chromatography/mass spectrometry (HPLC/MS) at 30 min, 1.5 h and 3 h and allowed to continue overnight to achieve complete conversion. Two batches displayed a very similar reaction profile and an ~85% final purity of the crude. Upon reaction completion, the two batches were combined in a round bottom flask to give a 5 mL solution of crude labeled bleomycin in DMSO, which was diluted with water (25 mL), snap-frozen and then freeze-dried overnight to remove the solvents. The resulting red solid was then triturated in dichloromethane (DCM) in order to remove impurities, notably succinimide-OH, which is released by the BODIPY-FL NHS upon coupling, and additional fluorescent byproducts. Next, 5 mL of DCM was added and incubated until all undissolved solid material settled at the bottom of the flask (approximately after 15 min). The supernatant containing impurities and side-products dissolved in DCM was removed by means of a syringe. This process was repeated five times until the supernatant was colorless. This final product was re-dissolved in water (10 mL), transferred in a vial, snap-frozen and then freeze-dried overnight again, resulting in 121 mg of a fine, red powder that was characterized by HPLC/MS. The final product was sent at 4 °C, in the dark, to be encapsulated into UD liposomes and formulated into a 0.2% cream (F-Bleosome). A small amount was retained as an equivalent non-encapsulated fluorescently labeled bleomycin (F-bleomycin).

### 2.5. Skin Sample Preparation

Canine and equine skin samples were collected from cadavers donated to the Royal (Dick) School of Veterinary Studies (RDSVS) at The University of Edinburgh with consent to be used for teaching and/or research. Specimens were taken from the shaved flank and abdomen of cadavers, and stretched taut by attaching to a cork block; subcutaneous fat was removed using a scalpel. Each sample was then immediately wrapped in aluminum foil, sealed in a plastic bag, snap-frozen and stored at −20 °C. The Veterinary Ethical Review Committee of The University of Edinburgh deemed that this study raised no ethical concerns.

### 2.6. Transmission Electron Microscopy

Skin samples were thawed at RT in PBS and cut to 2 × 2 cm^2^ sections. Each skin sample was treated with the indicated dose of drug for the indicated time. The appropriate sample was then cut into 1 cm thick slices that were fixed in 3% glutaraldehyde in 0.1 M sodium cacodylate buffer, pH 7.3 for 2 h then washed 3 × 10 min in 0.1 M sodium cacodylate buffer, pH 7.3. Samples were then post-fixed in 1% osmium tetroxide, 0.1 M sodium cacodylate for 45 min at RT, then washed 3 × 10 min in a 0.1 M sodium cacodylate buffer. Samples were then dehydrated in 50%, 70%, 90% and 100% ethanol for 3 × 15 min each followed by 2 × 10 min in propylene oxide and then embedded in TAAB812 resin. Ultrathin sections were cut using a Lecia Ultracut S ultramicrotome at a thickness of 60 nm, stained with 5% uranyl acetate and 2% lead citrate and imaged on a JEOL JEM-1400 TEM (JEOL, Ltd., Akishima, Japan) equipped with Gatan OneView camera (Gatan, Pleasanton, CA, USA) at King’s Buildings at The University of Edinburgh.

### 2.7. Multiphoton Microscopy

Skin samples were thawed at RT in PBS and cut to 2 × 2 cm^2^ sections. Each skin sample was treated with the indicated dose of drug for the indicated time. Each skin section was initially stained on both sides with Hoechst 33342 (Thermofisher Scientific) at 0.02 mg/mL in PBS by incubating for 30 min in the dark at RT on both sides. Skin samples were then washed in PBS and treated with the indicated dose of drug for the indicated time by spreading 0.1 mL cream evenly on the epidermal surface. Care was taken to avoid treating the sides of the section. After the indicated time point the residual cream was removed with a cotton swab and the sample was washed with PBS. All samples were positioned in 35 × 10 mm petri dishes (gridded Nunclon^TM^ Delta, Thermofisher Scientific) with the epidermis side anchored to the bottom of the dish by 4% agarose with the subcutaneous side facing upward, toward the microscope lens. The dish with the sample was filled with PBS and positioned in the visualising plate of the MP. All samples were imaged with a Zeiss (Cambridge, UK) LSM7 MP intravital multiphoton microscope equipped with W Plan-Apochromat 20×/1.0 DIC D = 0.17 M27 75 mm objective lens and a Coherent chameleon ultraII pulsed laser with OPO (wavelength 690 to 1400 nm). Green (NDD R4 ET535/30) and blue (NDD R3 ET460/36) filters were used. Multidimensional acquisition was performed with the Z-stack tool, setting 150 slices as average with an interval of approximately 3 μm. 3D images were visualised, processed and interpreted using Imaris (Biplane) software (version 9.1.2, Oxford Instruments, Abingdon, UK).

### 2.8. Quantification

Imaris 9.1.2 (Bitplane) software (Oxford Instruments, Abingdon, UK) was used to analyze 3D images acquired by MP. Structures labelled with Hoechst 33342 staining that did not emit or emitted a low green fluorescent signal (including a majority of auto-fluorescent skin appendages such as connective tissue) were considered as “surfaces” (“surfaces” had NDD R3-blue source channel), while all the elements emitting high levels of the green signal, either labelled with Hoechst 33342 (such as cellular nuclei) or not (F-Bleosome and F-bleomycin), were labelled as “spots” (“spots” had NDD R4-green source channel). After this distinction, the program distinguished between the “spots far from surfaces” (green particles emitting high- intensity green fluorescence far from the particles emitting blue and low-intensity green fluorescence), from the “spots close to surfaces” (green particles emitting high-intensity green fluorescence, and blue fluorescence very close to auto-fluorescent skin and cellular structures, emitting low-intensity green fluorescence), using a specific threshold (0.5). The aim was to target the green particles that were far from the auto-fluorescent skin structures and that were not stained with Hoechst 33342 (cellular nuclei were always close to intracellular auto-fluorescent elements and thus recognised as “spots close to surfaces”). This system enabled us to identify F-bleomycin and F-Bleosome particles within the examined skin section, recognised as green emitting particles far enough from auto-fluorescent structures (“spots far from surfaces”). All images were visualised from the subcutaneous side of the skin section. Imaris automatically compensated for the inversion and depths were compared to fluorescent images. In the equine skin samples, there was a discrepancy between the computer compensated values and the images, and here we manually compensated for this inversion by making all particles relative to the deepest particle in each image.

### 2.9. Statistics

Data were analysed using Minitab^®^ 17 Statistical Software (Minitab Ltd., Coventry, UK). *p*-values < 0.05 were considered statistically significant. When data followed a normal distribution, two-sample *t*-tests and Pearson’s chi-squared tests were used to compare differences between two samples, or one-sample *t*-tests to determine whether the sample mean was statistically different from a known or hypothesised mean. A general linear model (2-way ANOVA- analysis of variance) was used to understand if in two groups there was an interaction between two independent variables on the dependent variable. A Fisher’s post-hoc test was used to compare differences between time points post-treatment.

## 3. Results

### 3.1. Preparation of BODIPY-Labeled Bleomycin for Fluorescent Imaging of Bleosome through the Skin

In this study we were assessing if a novel formulation of bleomycin encapsulated in UD liposomes penetrates through the outer layers of the skin better than the free drug. Fluorescent labeling of a drug is a valuable tool to allow visualization within an enclosed biological system, and in this case to quantify the degree of penetration through the skin. When preparing fluorescent versions of drugs, labels must be introduced at appropriate amounts and positions so they do not interfere with activity. Here, we aimed to produce mono-labeled bleomycin as we considered that this would be less disruptive to the pharmacological activity of the drug than the incorporating of multiple labels (Figure 1A).

Preliminary experiments were carried out to determine the optimum fluorophore to bleomycin ratio to produce mono-labeled bleomycin. The experimental procedure is detailed in materials and methods and highlighted in Figure 1B. The coupling reaction was monitored by HPLC/MS (Figure 1C). The final compound contained 88% mono-labeled and 12% free-bleomycin (Figure 1C(vi)), and was subsequently encapsulated into UD liposomes to produce fluorescent Bleosome (F-Bleosome) and the equivalent un-encapsulated control (F-bleomycin).

### 3.2. F-Bleosome Retains the Same Efficacy as Unlabeled Bleosome

To determine if the coupling of BODIPY-FL to bleomycin impaired the cytotoxicity of the drug, a canine mastocytoma cell line (C2) and a canine melanoma cell line (CML10) were treated with increasing doses of either F-Bleosome or Bleosome and cell viability was assayed 48 h after treatment (Figure 1D). There was no significant difference between the killing effect of F-Bleosome and Bleosome, indicating that the fluorophore coupled to bleomycin did not affect the cytotoxicity of the drug. These results were consistent in both cell lines tested.

### 3.3. Liposomes Do Not Penetrate beyond the Stratum Corneum

Canine skin explants were treated with 0.1 mL of Bleosome for 0 (vehicle control), 0.5, 2, 4, 6 and 8 h, fixed and visualized by TEM. All the epidermis was visible including the layers of the stratum corneum. Within the 0 h control, no liposomes were observed in any of the epidermal structures (Figure 2A), whereas liposomes were visible in all skin explants treated with Bleosome over the time course (Figure 2B–F). Liposomes do penetrate through the corneocytes of the stratum corneum but do not penetrate deeper over time and were not identified beyond the external keratinocyte layer (Figure 2B–F).

### 3.4. Bleosome Penetrates Further through Skin Than Bleomycin

Canine and equine skin sections were treated with 0.1 mL of either F-Bleosome or F-bleomycin for 0 h, 10 min, 4 h and 6 h and visualized by MP. The skin has several structures that are inherently auto-fluorescent such as collagen, hair follicles and cellular nuclei. All images were compared to the 0 h control to establish background fluorescence. Hoechst staining was used to stain nuclei blue, however the intensity of the blue was similar to the green of BODIPY-FL, to differentiate we set the ‘blue’ of Hoechst 33342 to ‘magenta’ during image acquisition. At the image analysis stage, in the merged image, when magenta and green overlap it appears as white. Hoechst 33342 does not stain liposomes or bleomycin, therefore objects appearing green are considered as the drug.

Treated canine skin samples are shown from the side-view and the top-view in Figure 3A,B, respectively. The untreated control did not display any distinct solid green particles but there was a basal level of auto-fluorescence, mainly due to cutaneous structures, that were also stained by Hoechst and appeared white on the merged image (Figure 3A(i),B(i)). Distinct round green fluorescent dots, corresponding to fluorescent BODIPY-FL-labeled bleomycin encapsulated in UD liposomes were visible at a superficial level in the skin sample 10 min after treatment with F-Bleosome (Figure 3A(ii),B(ii)). Over time F-Bleosome particles penetrate deeper through the skin, at a higher overall number and increased in size with a less defined spherical shape (Figure 3A(iv,vi,vii),B(iv,vi,vii). In comparison, the fluorescently labelled free drug, F-bleomycin, is not readily detectable in the deeper layers of the skin and there is no notable change over treatment time of number, size or depth of F-bleomycin (Figure 3A(iii,v,vii),B(iii,v,vii). Imaris software was used to identify and quantify the number of fluorescent particles within each sample. Calibration of the software, as detailed in materials and methods, failed to identify the fluorescently labelled drug located close to auto-fluorescent skin structures, leading to a consistent underestimation of the overall number of fluorescent particles. Skin sections treated with F-Bleosome contained a significantly higher number of particles at all time points than those treated with F-bleomycin (Table 1, *p*-value 0.002). Ten minutes post-treatment F-Bleosome and F-bleomycin particles were found at a similar superficial level within the skin (Figure 3C(i)), but by 4 h and 6 h post-treatment F-Bleosome particles were identified significantly deeper (at an average depth of 307 μm and 452 μm, respectively) than F-bleomycin (at an average depth of 167 μm and 237 μm, respectively) (Figure 3C).

Equine skin showed similar results to canine skin whereby more F-Bleosome particles penetrated through the skin and to a greater depth than F-bleomycin at the indicated time points (Figure 4A,B). Using Imaris software we determined that the overall number of particles was significantly higher at each time point for those skin samples treated with F-Bleosome compared to F-bleomycin (Table 2, *p*-value 0.000) and significantly deeper at 4 h and 6 h post-treatment (at an average depth of 117 μm and 166 μm, respectively, for F-Bleosome compared to 67 μm and 65 μm, respectively, for F-bleomycin) (Figure 4C). Our results confirm that UD liposomes are effective drug carriers of bleomycin through the canine and equine skin.

## 4. Discussion

Bleomycin is an anti-neoplastic antibiotic used in treatment regimens to treat several types of human cancer including cervical and uterine cancer, squamous cell carcinoma, testicular and penile cancer and certain types of lymphoma [22,23,24]. The clinical use of bleomycin as a stand-alone therapy is limited by its toxicity to the lung causing irreversible pulmonary damage and fibrosis in approximately one fifth of all patients [25]. However, bleomycin has potential as a topical treatment in both human and veterinary medicine for diseases such as NMSCs and equine sarcoids, due to its low toxicity against non-cancer cells and hematopoietic tissues [26,27], however its molecular structure impedes penetration through the skin [4]. The application of nanotechnology to overcome biological hurdles is overhauling biomedical research and clinical applications. Here, we utilized a novel formulation of bleomycin encapsulated in UD liposomes (Bleosome).

Bleosome has previously been shown to have a lethal effect on an immortalized human keratinocyte cell line and a cell line derived from a primary squamous cell carcinoma [13] and has potential as a clinically relevant modality for horses with occult and/or verrucose sarcoids [28]. However, to date there has been limited evidence that Bleosome can effectively penetrate through the skin compared to free-bleomycin and limited insight into the mechanism of the UD liposomes’ enhanced skin penetration. In this study, we showed for the first time that Bleosome effectively penetrates deeper into the skin than free-bleomycin, and that these results are consistent in two model systems: canine and equine. We visualized UD liposomes approximately 150 nm in diameter using TEM. TEMs are powerful analytical tools that transmit a beam of electrons through a thin slice of a tissue to capture the very fine details of that specimen; they are of a magnification of up to thousands of times higher than that of a light microscope [29]. UD liposomes did not penetrate beyond the external keratinocyte layer of canine skin even after 8 h of treatment. Similar results were observed by Bouwstra et al. (2003) [30], whereby intact surfactant-based elastic vesicles were not found beyond the stratum corneum of human skin.

To complement the TEM, we fluorescently labelled bleomycin prior to encapsulation in UD liposomes to enable us to visualize the drug by MP (Figure 5A). BODIPY-FL was selected, as it is relatively non-polar and non-charged fluorophore that should have minimum effect on the functional properties of bleomycin after conjugation [31]. To our knowledge this is the first time that BODIPY-FL has been conjugated to bleomycin, and our main concern was that the reactive succinimidyl group of BODIPY-FL NHS could potentially bind to all available amine groups in the metal binding domain of bleomycin, which may impair the DNA binding and cleavage and affect the efficacy of the fluorescently labeled drug [4]. We effectively mono-labeled bleomycin with BODIPY-FL prior to encapsulation in UD liposomes (F-Bleosome) and confirmed the comparable cytotoxic effect on cell lines compared to unlabeled Bleosome. Using F-Bleosome we were able to visualize the drug within the UD liposomes as it penetrated through the skin. We found that F-Bleosome penetrated beyond the keratinocyte layer and penetrated deeper over time compared to F-bleomycin, which was mainly retained in the superficial layers of the skin. The epidermal thickness of the skin is estimated to be approximately 40 μm and 30 μm for canine [32] and equine [33] skin respectively, and in both models F-Bleosome penetrated beyond the epidermis and was identified within the dermal layers of the skin. In some instances we also identified F-Bleosome particles in close proximity to structures that are likely to be small blood vessels, indicating that F-Bleosome can penetrate beyond the epidermis, an avascular component of the skin. We noted that the shape of F-Bleosome particles changed over time: 10 min post-treatment they are well demarcated and spheroidal whereas at later time points there were less clearly defined and more diffuse. Given that UD liposomes, as visualized by TEM, were not observed beyond the stratum corneum, we propose that UD liposomes act as penetration enhancers. Our model is consistent with that proposed by Verma et al. (2003) [20], whereby within the superficial layers of the skin Bleosome consists of bleomycin encapsulated within UD liposomes; once Bleosome passes beyond the stratum corneum, we hypothesize that the UD liposomes are damaged by their interaction with hardy keratinocytes, which are highly keratinized and flat, causing disruption of the liposomal membrane. These damaged UD liposomes then become leaky, and the previously encapsulated bleomycin is released and able to freely penetrate deeper into the cellular and vascular dermal layer of the skin (Figure 5B). We therefore propose that UD liposomes are penetration enhancers that can transport encapsulated drugs across the epidermis and release them in the deeper layers of the skin.

Our results support further studies to investigate the clinical application of Bleosome as a topical treatment for NMSCs. In both human and veterinary patients these types of tumors are usually not fatal, but their treatment represents a clinical challenge as they are often highly invasive and result in life-long disfiguration of the patient [1]. An effective topical treatment for these types of tumors would negate the need for patients to undergo disfiguring surgery and could be safely applied at home. Clinical use in human NMSCs has shown minimal side-effects and scarring (Chopra, S. personal communication). Within the context of veterinary medicine, topical treatments should be designed for rapid absorption to minimize companion animals from licking and swallowing any external treatments, which may be harmful if swallowed. Here we showed that Bleosome can more effectively penetrate through the healthy skin of dogs and horses, and can reach the deeper layers of skin better than the free drug. Future studies should focus on the rate of absorption, penetration through tumor samples, and efficacy against NMSCs. Two promising preliminary studies showed that Bleosome in combination with either 5-fluorouracil or tazarotene was more effective at treating horses with sarcoids than either treatment alone [28], and Bleosome alone has been used to treat humans with NMSC (Chopra, S. personal communication). The current study adds to a body of evidence that Bleosome is potentially an effective non-invasive treatment, with minimum side-effects, for NMSC in both human and veterinary medicine.

## 5. Conclusions

Innovation and novel formulations of established anti-neoplastic drugs have the potential to affect clinical practice and patient outcomes quicker than completely new drugs for which there is no prevailing pharmacological, pharmacokinetic or clinical data available. Bleosome is a novel formulation of bleomycin that has potential as a topical chemotherapy to treat NMSC and negate the adverse side-effects of pulmonary toxicity when administered systemically. In this study we show for the first time that Bleosome can effectively penetrate through the skin of dogs and horses better than the free drug which cannot pass beyond the stratum corneum. Using novel imaging techniques, we have also elucidated the mechanism of UD liposome enhanced skin penetration of bleomycin. From our results we conclude that Bleosome is able to penetrate through skin better than free-bleomycin as UD liposomes can penetrate through the protective, hardy, outer layers of skin which impair passage of the free-drug, and then once beyond the keratinocyte layer the UD liposomes break-up to release the drug, enabling it to diffuse deeper through the skin. This study provides a scientific underpinning to preliminary clinical observations that Bleosome may be an effective treatment of NMSCs.

## 6. Patents

S.C holds a patent (GB2398495) “A drug delivery preparation comprising at least one anti-tumor drug and a topical carrier for the drug” and has a patent pending for “Compounds, Compositions and Methods for the Treatment or Prevention of Hair Loss”.

## Figures and Tables

**Figure 1 cancers-14-01083-f001:**
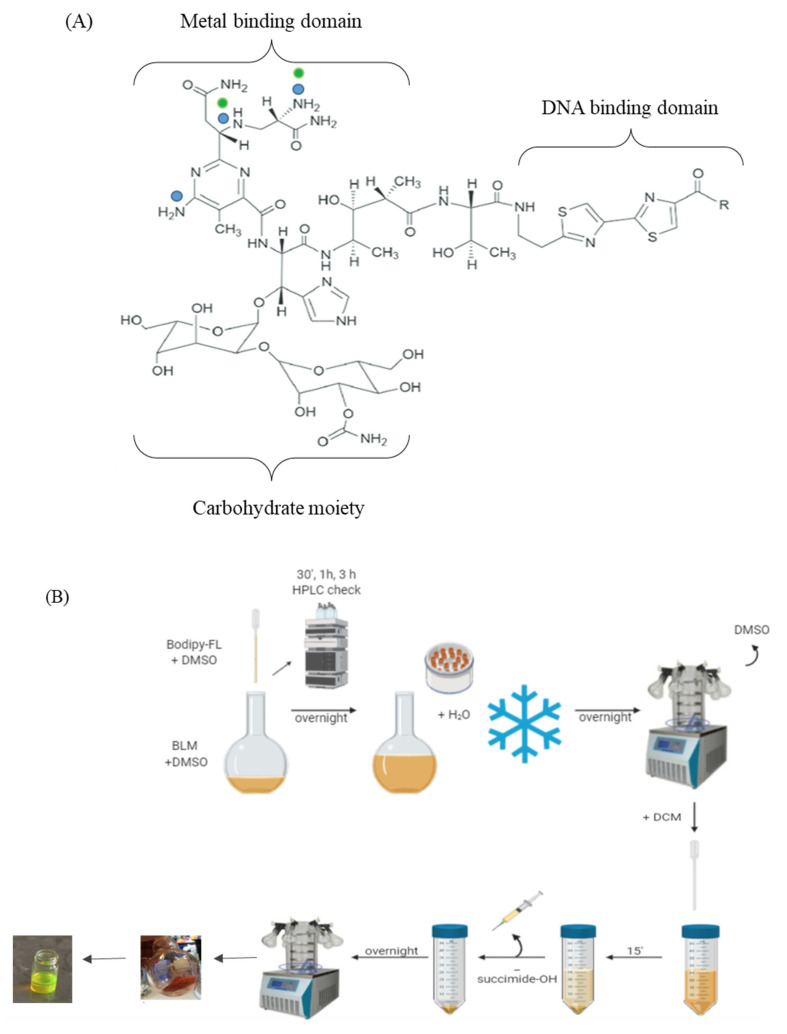
Fluorescent labeling of bleomycin with BODIPY-FL. (**A**) Molecular structure of bleomycin showing the metal binding domain with the potential amine groups available for BODIPY-FL binding highlighted (dots: blue dots indicate all potential binding sites; and green dots indicate preferred binding sites); the DNA binding domain encompassing the bithiazole tail; and the carbohydrate moiety required for cellular uptake. (**B**) Schematic representation of the labeling process of bleomycin with BODIPY-FL. The reaction products of BODIPY-FL NHS added to bleomycin sulfate were analyzed by HPLC/MS and recovered from DMSO by lyophilization. Succinimide-OH and fluorescent impurities were removed by trituration with DCM. The compound was dried overnight and the final product obtained was a red powder (which turns fluorescent green in solution). (**C**) HPLC profiles of coupling of BODIPY-FL to bleomycin. Using 45 mg bleomycin after (**i**) 30 min and (**ii**) overnight. Using 71 mg of bleomycin after (**iii**) 30 min and (**iv**) overnight. Combined labelled bleomycin batches (**v**) before and (**vi**) after the final lyophilization step. According to their absorbance profile and mass spectra: peak 1.83 (red circle) corresponds to succinimide-OH; peak 3.38 (black circle) to free bleomycin; peak 4.22–4.12 (green circle) corresponds to the mono-labeled bleomycin; 5.20 (yellow circle) corresponds to double-labeled bleomycin; peak 7.63 (blue circle) corresponds to free (unbound) BODIPY-FL; 8.05 (violet circle) corresponds to a minor impurity of the fluorophore. (**vi**) the final compound consisting of 87.55% mono-labeled bleomycin and 12.45% free bleomycin. (**D**) F-Bleosome has comparable cytotoxic effects to unlabeled Bleosome. The cancer cell lines C2 (**i**) and CML10 (**ii**) cells were treated with the indicated concentrations of Bleosome and F-Bleosome, and cell viability was assayed after 48 h.

**Figure 2 cancers-14-01083-f002:**
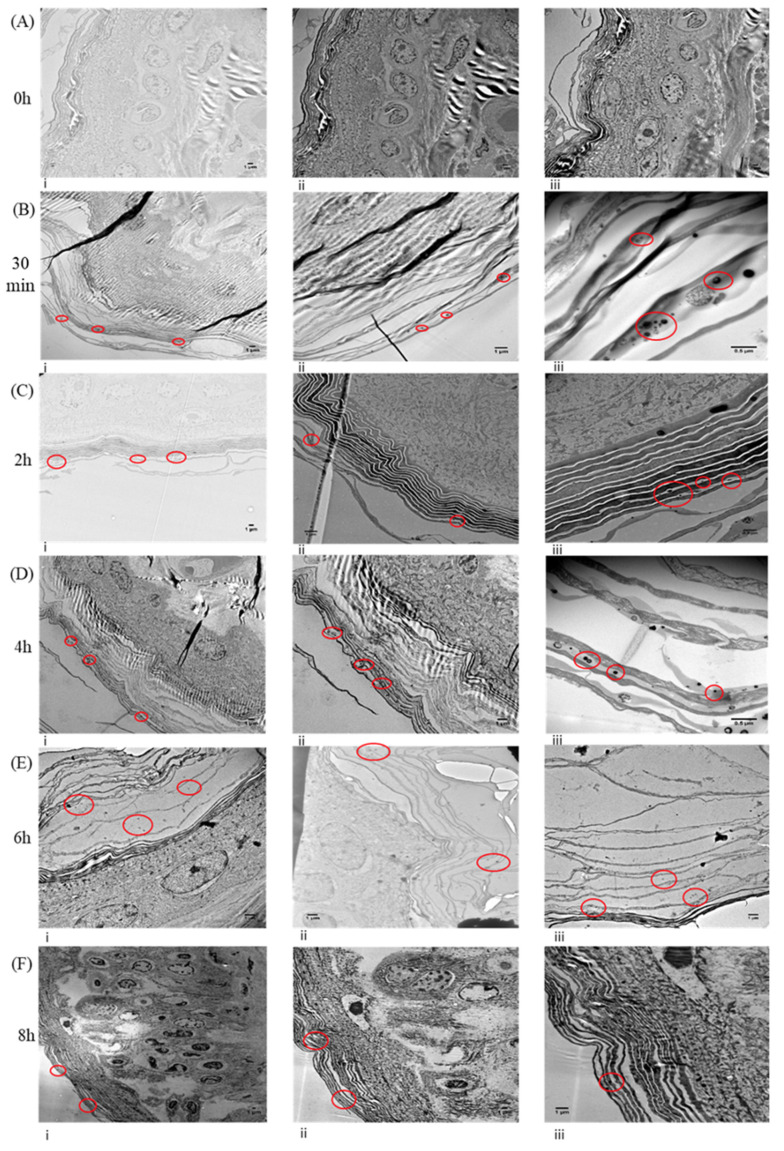
Transmission electron microscopy visualization of liposomes through canine skin at different time points after a single Bleosome administration. (**A**) Skin samples were treated for 0 h (control) with Bleosome, visualized at different magnification: (**i**) ×1200, (**ii**) ×1200, (**iii**) ×1200; (**B**) images of samples treated for 30 min with Bleosome: (**i**) ×1200, (**ii**) ×5000, (**iii**) ×10K; (**C**) images of samples treated for 2 h: (**i**) ×500, (**ii**) ×2500, (**iii**) ×5000; (**D**) images of samples treated for 4 h: (**i**) ×1200, (**ii**) ×2500 K, (**iii**) ×10 K; (**E**) images of samples treated for 6 h: (**i**) ×2500, (**ii**) ×2500, (**iii**) ×2500; (**F**) images of skin samples after 8 h of Bleosome treatment: (**i**) ×500, (**ii**) ×1200, (**iii**) ×2500. Representative liposomal nanoparticles are circled in red within the keratinocytes of the stratum corneum only.

**Figure 3 cancers-14-01083-f003:**
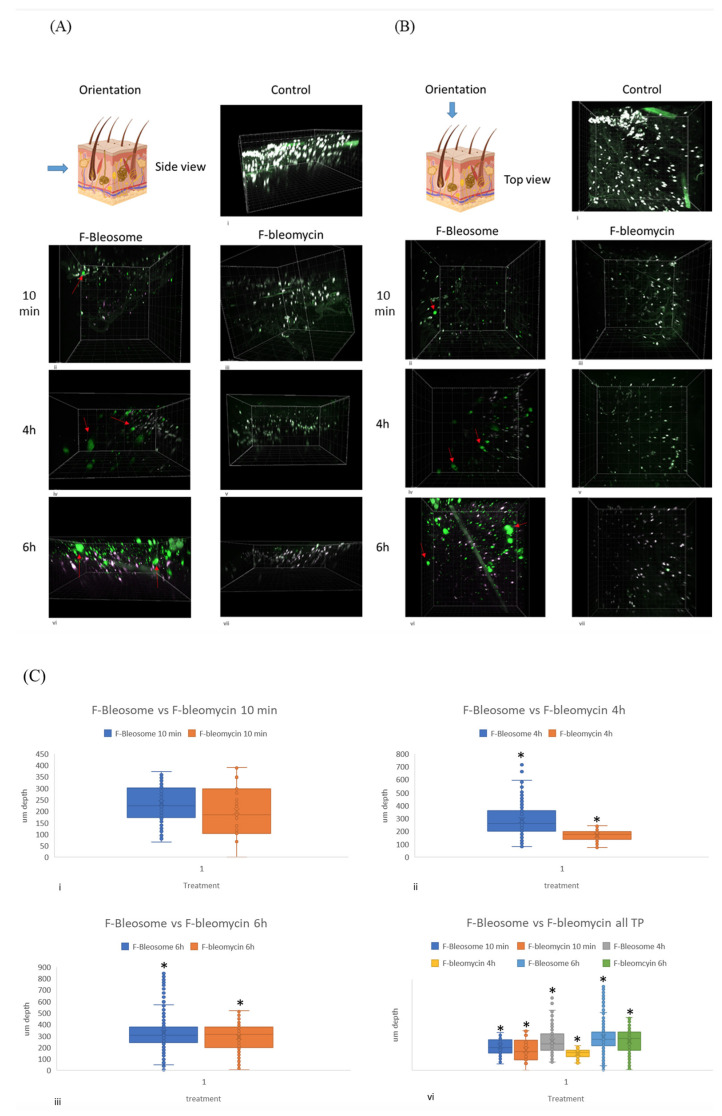
Bleosome penetrates deeper into canine skin than free-bleomycin. (**A**) Side view of canine skin sections treated with F-Bleosome and F-bleomycin at the indicated time points. (**i**) control (untreated skin section); (**ii**,**iv**,**vi**) correspond to the 3-D images of the skin sections treated with F-Bleosome after 10 min, 4 h and 6 h. Representative F-Bleosome molecules are indicated by red arrows. (**iii**,**v**,**vii**) correspond to the side view of the 3-D sections after 10 min, 4 h and 6 h of F-bleomycin treatment. (**B**) Top view of corresponding canine skin sections. (**i**) control (untreated skin section); (**ii**,**iv**,**vi**) correspond to the 3-D images of the skin sections treated with F-Bleosome after 10 min, 4 h and 6 h. Representative F-Bleosome molecules are indicated by red arrows. (**iii**,**v**,**vii**) correspond to the side view of the 3-D sections after 10 min, 4 h and 6 h of F-bleomycin treatment. (**C**) Box plots comparing the depth of the drug molecules (µm) through the canine skin sections treated with F-Bleosome (blue plot) and F-bleomycin (orange plot) after (**i**) 10 min, (**ii**) 4 h and (**iii**) 6 h. (**iv**) Comparison of all three time points. * Indicates statistically significant difference (*p*-value < 0.05, two-way ANOVA).

**Figure 4 cancers-14-01083-f004:**
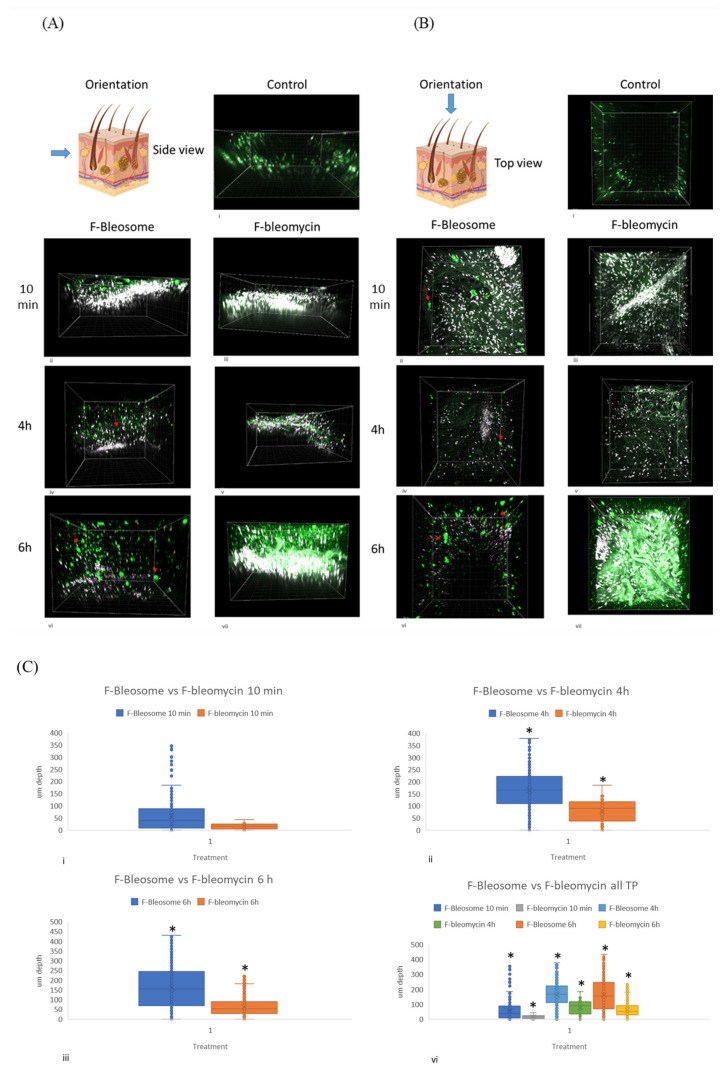
Bleosome penetrates deeper into equine skin than free-bleomycin. (**A**) Side view of equine skin sections treated with F-Bleosome and F-bleomycin at the indicated time points. (**i**) control (untreated skin section); (**ii**,**iv**,**vi**) correspond to the 3-D images of the skin sections treated with F-Bleosome after 10 min, 4 h and 6 h. Representative F-Bleosome molecules are indicated by red arrows. (**iii**,**v**,**vii**) correspond to the side view of the 3-D sections after 10 min, 4 h and 6 h of F-bleomycin treatment. (**B**) Top view of corresponding equine skin sections. (**i**) control (untreated skin section); (**ii**,**iv**,**vi**) correspond to the 3-D images of the skin sections treated with F-Bleosome after 10 min, 4 h and 6 h. Representative F-Bleosome molecules are indicated by red arrows. (**iii**,**v**,**vii**) correspond to the side view of the 3-D sections after 10 min, 4 h and 6 h of F-bleomycin treatment. (**C**) Box plots comparing the depth of the drug molecules (µm) through the equine skin sections treated with F-Bleosome (blue plot) and F-bleomycin (orange plot) after (**i**) 10 min, (**ii**) 4 h and (**iii**) 6 h. (**iv**) Comparison of all three time points. * Indicates statistically significant difference (*p*-value < 0.05, two-way ANOVA).

**Figure 5 cancers-14-01083-f005:**
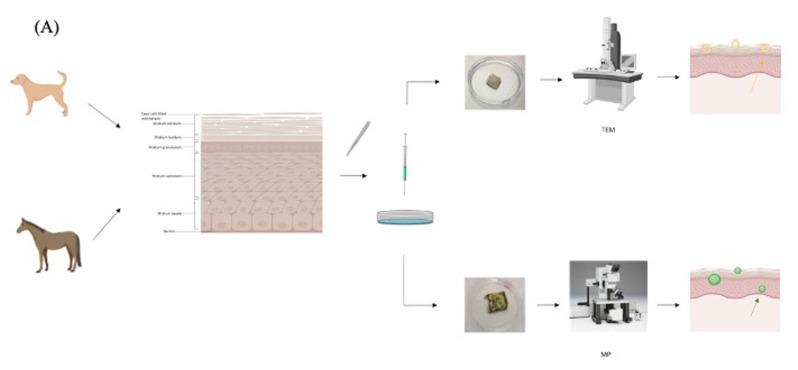
Proposed mechanism of penetration of Bleosome through the skin. (**A**) Schematic representation of the experimental techniques used to assess the penetration of Bleosome through the skin. Skin was collected from cadavers of dogs and horses. Subcutaneous fat was removed using a scalpel and hair (in the case of dogs and horses) was removed using electric clippers. Skin explants were cut into 2 × 2 cm^2^ sections and treated with either Bleosome or free bleomycin and visualized by the TEM (top) for liposomes assessment (yellow arrow) or treated either with F-Bleosome or F-bleomycin and visualized by the MP (bottom) for drug particles assessment (green arrow). (**B**) UD liposomes act as penetration enhancers to increase the penetration of encapsulated bleomycin through animal and human skin. We propose that when liposomal vesicles interact with the outermost keratinocytes, they are disrupted, allowing the release of the entrapped bleomycin and consequently the enhanced penetration of the drug to the inner layers of the skin.

**Table 1 cancers-14-01083-t001:** Number of particles detected within canine skin treated with either F-Bleosome of F-bleomycin after 10 min, 4 h and 6 h.

Time Point	Treatment	Average Number of Particles (*n* = 3)
10 min	F-Bleosome	87
4 h	F-Bleosome	85
6 h	F-Bleosome	618
10 min	F-bleomycin	8
4 h	F-bleomycin	34
6 h	F-bleomycin	178

**Table 2 cancers-14-01083-t002:** Number of particles detected within equine skin treated with either F-Bleosome of F-bleomycin after 10 min, 4 h and 6 h.

Time Point	Treatment	Average Number of Particles (*n* = 3)
10 min	F-Bleosome	120
4 h	F-Bleosome	305
6 h	F-Bleosome	500
10 min	F-bleomycin	4
4 h	F-bleomycin	37
6 h	F-bleomycin	412

## Data Availability

The data presented in this study are available in article.

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
