# Peer review of "Effective Penetration of a Liposomal Formulation of Bleomycin through Ex-Vivo Skin Explants from Two Different Species"

_cancers, 2022, doi:10.3390/cancers14041083_

Round 1

Reviewer 1 Report

The manuscript presents innovative technology based on well-chosen research methods. The conclusions are of great practical importance.

The structure of the manuscript is correct, the content is understandable to the reader. The authors draw the right conclusions from the work on research.

The main question addressed by the research - The authors present the results of their analysis devoted to the new formulation of bleomycin encapsulated in ultra-deformable liposomes. The topic of the manuscript is original and novelty. The results have an important influence on clinical practice.  The important advance of this manuscript is detailed comparative analysis of two formulation of bleomycin in vivo and in vitro. The methodology of the analysis is adequate. The conclusions are consistent with the evidence and arguments presented and they address the main question posed. The tables and figures are necessary and allow to better understand the text for readers. The references are appropriate. 

Author Response

Thank you for reviewing our paper. Please find attached our response to your comments. Many thanks. 

Reviewer 2 Report

This interesting study demonstrates by imaging analysis that bleomycin is able to penetrate the skin layers via an UD liposome formulation. Bleomycin is a large hydrophilic molecule with polar charge. It cannot penetrate the skin without any carrier. Thus, it is a systemic anticancer drug. Bleosome is an ultra-deformable (UD) liposome, which has been shown with ability to inhibit skin cancer when being applied topically in previous studies. The strength of the study includes a unique fluorescently labeling bleomycin with BODIPY in order to visualize the penetration of both liposome (visualized by TEM) and the drug (imaged by multiphoton microscopy). The image analysis demonstrates that the liposome part stays in the SC layer, does not get into the deep layers of the skin, while the drug encapsulated in the liposome was able to go deeper. This finding indicates that UD liposome may act as a penetration enhancer. However, bleosome, which is the UD liposome formulation, has been published in 2005 (reference #13). Thus, the title “a novel formulation of bleomycin” is misleading since this study did not create a novel formulation but conduct mechanistic study to figure out how bleosome penetrates the skin. Another limitation is that this study does not include any efficacy and toxicity data, which fails to provide any evidence for the stated notion that topical bleomycin shows limited side effects (line 26). This study is in vitro (ex vivo) only. Furthermore, the study conclusion should be supported independently by drug penetration data using Franz diffusion system and HPLC (or LC/MS) analysis. In addition, the reviewer feels that this study may be more suitable for journals in the drug delivery field.

Minor comments:

Line 22: extra “a”

Fig 1D should test F-bleomycin vs. unlabeled fleomycin because bleosome shows limited cytotoxicity on two selected cell lines. Fig1Dii should change the Y- scale.

Author Response

(The authors gave the same response as above.)

Reviewer 3 Report

It's an interesting study that suggests that a novel formulation of bleomycin encapsulated in UD liposomes penetrates through the outer layers of the skin better than the free drug and may be accepted after major revision.

  1. Line 118, correction its CO2 incubator
  2. Cell density, Is it 500 cells/well or 5000 cells/well, please check and insert the correct number.
  3. Figure 2, images are not clear, not to be improved such B
  4. References should be updated for the year 2020 and 2021

Author Response

(The authors gave the same response as above.)

Round 2

Reviewer 2 Report

The authors answered my questions. 

Author Response

Thank you for reviewing our paper. I am pleased that we could address all of your concerns. 

Reviewer 3 Report

Regarding the previous question about the number of cells seeded in 96 well plates, they have mentioned 500 cells/well, it's too low a number of cells to be seeded, seeding 500 cells/well, will take many days to confluence.  I  disagreed with the authors, if they have used 500 cells/well, they need to support published evidence to support such claim. It appears that cells were gifted from Dr. Lauren Wolfe, Auburn University. I need to know the source of purchase from cell lines. They used two cell lines (1)  canine melanoma cell line, CML10 (2) A canine mastocytoma cell line, C2. Are both of these cells gifted by Dr. Lauren, please clarify. I would also like to know the passage number of each cell line which were used for testing.  

Author Response

Thank you for reviewing our paper. Please find attached our response to your comments highlighted in red. Many thanks. 

Round 3

Reviewer 3 Report

The authors have satisfactorily responded to queries and the revised manuscript has been improved and may be accepted for publication.